# Cleft Palate and Aortic Dilatation as Clues for Loeys–Dietz Syndrome

**DOI:** 10.3390/children9091290

**Published:** 2022-08-26

**Authors:** Pierluigi Zaza, Flavia Indrio, Annalisa Fracchiolla, Matteo Rinaldi, Giovanni Meliota, Alessia Salatto, Antonio Bonacaro, Gianfranco Maffei

**Affiliations:** 1Ospedali Riuniti Foggia Italy-U.O.C. Neonatologia e Terapia Intensiva Neonatale, 71100 Foggia, Italy; 2Department of Medical and Surgical Science Pediatric Section, University of Foggia, 71100 Foggia, Italy; 3Ospedale Giovanni XXIII U.O. Cardiologia Pediatrica, 70124 Bari, Italy; 4DAI Materno-Infantile, Azienda Ospedaliera-Universitaria Federico II di Napoli, 80100 Napoli, Italy; 5School of Health and Sports Sciences, University of Suffolk, Ipswich IP4 1QJ, UK

**Keywords:** Loeys–Dietz syndrome, cleft palate, aortic dilatation, genetic syndrome

## Abstract

Loeys–Dietz syndrome (LDS) is a rare autosomal-dominant disorder of the connective tissue with some typical vascular findings, skeletal manifestations, craniofacial features, and cutaneous findings with a wide phenotypic spectrum. Six different genes are involved in LDS and the diagnosis is based on the identification of a heterozygous pathogenic variant in *TGFBR1*, *TGFBR2*, *SMAD3*, *TGFB2*, *TGFB3*, or *SMAD2* in children with suggestive findings. These genes distinguish LDS into six classes (LDS1–LDS6, respectively). Delay in diagnosis of Loeys–Dietz syndrome may be associated with an adverse prognosis due to a very high augmented risk of early complications such as aortic or vascular rupture. The present report describes a case of an early diagnosis of LDS in a neonate with cleft soft palate and aortic root dilatation.

## 1. Introduction

Bart Loeys and Harry Dietz first described the Loeys–Dietz syndrome due to mutations in the *TGFBR1* and *TGFBR2* genes in 2005 in 10 patients who showed novel aortic aneurysm syndrome and a clinical triad of hypertelorism, bifid uvula/cleft palate and aortic/arterial aneurysms, and tortuosity [1]. Further features were also recognized, defining LDS as a combination of vascular findings (cerebral, thoracic, and abdominal arterial aneurysms and/or dissections), skeletal manifestations (pectus excavatum or pectus carinatum, scoliosis, joint laxity, arachnodactyly, talipes equinovarus, cervical spine malformation and/or instability), craniofacial features (widely spaced eyes, strabismus, bifid uvula/cleft palate, and craniosynostosis that can involve any sutures), and cutaneous findings (velvety and translucent skin, easy bruising, and dystrophic scars) [2]. Wide variation in the distribution and severity of clinical features can be seen in individuals with LDS as a different involvement of connective tissues has been observed in this syndrome, ranging from a precocious syndromic presentation in newborns and toddlers, with typical facial dysmorphisms and severe systemic features, to isolated aortic aneurysms incidentally discovered in adults. Cardiovascular manifestations are the leading causes of morbidity and mortality in LDS patients. Patients generally present with dilation of the aorta and pulmonary arteries, and therefore have a predisposition for aortic dissection. At a first evaluation, this syndrome can be misdiagnosed as the well-known Marfan syndrome (MS), which shares some clinical features. Yet, patients seem to be not particularly tall and have no lens dislocation, and clinical progression of aortic aneurysms or dissection in patients with LDS is faster with no effective catheter-based treatment. LDS is caused by heterozygous mutations in six different genes, which encode components of the transforming growth factor beta (*TGFβ*) pathway [3].

The diagnosis of LDS is established on characteristic clinical findings in the proband and family members and/or by the identification of a heterozygous pathogenic variant in *TGFBR1*, *TGFBR2*, *SMAD3*, *TGFB2*, *TGFB3*, or *SMAD2* encoding for components cooperating within the *TGFbeta* receptor signaling cascade to regulate cell differentiation and the development of many tissues, including bone, cartilage, blood vessels, and the heart. Loss-of-function heterozygous variants in *TGFBR1* and *TGFBR2* are the most frequent in LDS patients. Each gene is associated with different clinical characteristics that define a particular genotype–phenotype model. The clinical picture defined by mutations in the *TGFBR1*–*TGFBR2* genes is largely overlapping in the majority of LDS cases. Patients with this mutation frequently have an ascending aortic root aneurysm and severe arterial involvement. Valve anomalies, in particular, prolapse and insufficiency of the mitral valve, are rarer. Craniofacial malformations are significant especially in patients with the *TGFBR1* (LDS1) mutation including dolichocephaly, hypertelorism, malar hypoplasia, and a strongly arched palate, while abnormal palate and uvula bifida are not as frequent. LDS3 is instead defined by mutations in the *SMAD3* gene, and what distinguishes it from other classes of LDS is the finding of arthrosis in the patient with this mutation and less cardiovascular involvement [4]. More rarely, the *TGFB2* gene mutation defined as LDS class 4 is characterized by mild cardiovascular anomalies and a more important skeletal involvement with joint hypermobility, pectoral deformity, and scoliosis. Recently this genetic subtype has also shown instability of the cervical spine and ectopia lentis. At the moment, there is still little information in the literature about *TGFB3* and *SMAD2* mutations. Research on these genotype–phenotype models is still open, but we can confirm that patients with these mutations share much of the cardiovascular and connective tissue characteristics common to LDS.

LDS is an autosomal-dominant disorder with both non-penetrance and mosaicism reported. Approximately 25% of individuals diagnosed with LDS have an affected parent; the remainder of probands have LDS due to a de novo pathogenic variant. In general, the more severe cases with marked craniofacial and skeletal findings result from a de novo mutation, whereas the milder cases tend to be familial. Prenatal diagnosis for pregnancies at increased risk for LDS is possible if the pathogenic variant in the family is known. Ultrasound examination in the first two trimesters has a low sensitivity in detecting manifestations of LDS, but prenatal occurrence of aortic dilatation can be detected in the more striking cases. LDS has become a focus of researchers worldwide, leading to an improved understanding of the disease; however, LDS remains insufficiently studied. Currently, no standard treatment or guidelines for LDS patients exist, and few studies have been conducted. What is known is derived from other connective tissue diseases, including Marfan syndrome. Various medical approaches have been studied to slow down the development of vascular abnormalities in these genetic conditions, including beta blockers, ACE inhibitors, and angiotensin receptor blockers (ARBs), especially losartan. This report presents the findings of a genetically confirmed LDS case during the neonatal age.

## 2. Case Report

Our proband was the second child of healthy and non-consanguineous parents. The mother, a 33-year-old smoker of about ten cigarettes a day, had another healthy 4-year-old child and had a spontaneous abortion. No remarkable disease was recorded in the family history. Neither the prenatal blood exams nor ultrasonography evaluation showed any problem. The proband was delivered by vaginal birth at our hospital at 39 weeks and 5 days of gestation after an uneventful pregnancy. He had normal adaptation to the post-natal life; the Apgar score was 8 and 9, respectively, at the 1st and 5th minute after birth. During the first inspection of the oral cavity in the neonatal resuscitation unit, the neonatologist found a cleft soft palate (Figure 1), so the neonate was admitted for further examination. His birth weight was 3520 kg (+0.05 SDS on Italian neonatal growth charts), length 48 cm (−1.52 SDS), and head circumference 37 cm (+1.93 SDS). A more detailed examination showed other suggestive findings including arachnodactyly, feet joints stiffness, peripheral joint laxity, and slightly spaced eyes. No signs of congenital hip dysplasia emerged from the physical screening tests. The cardiovascular and thoracic examination were normal; the abdomen did not show any pathological features; external genitalia appeared of male type with both testicles palpable in the scrotal sac; and the anus was patent and normally set. No cardiopulmonary problems were observed during the hospitalization, with normal peripheral oximetry and normal heart rate for age. The newborn had an acceptable suction reflex and enteral feeding through a bottle was possible from the first day of life with special cleft feeders. A transient-evoked otoacoustic emission hearing test was not passed either in the right or left hear. The complete blood count, liver and kidney function markers, and inflammation markers were all normal at the fourth day. The cranial ultrasound performed in the 3rd day showed multiple cysts in the frontal horn of both the lateral ventricle and other germinolytic cysts in the subependymal area (Figure 2). The kidney ultrasound appearance and hip ultrasound screening were normal. Regarding the abdominal organs, an enlargement of 9 mm of the left branch of the portal vein with a slow flow in the color-Doppler investigation was evident in the ultrasound.

The more striking results were obtained at the bedside echocardiography, which showed an aneurysm of the aortic root (16 mm; z-score +7.46) with a mild regurgitation of the aortic valve, a normal ascending aorta (10 mm; z-score +1.87) and a tortuous aortic arch with patent ductus arteriosus. A chest x-ray was performed to rule out other cardiopulmonary anomalies, but it documented just a slight augmented cardiac diameter. During a pediatric cardiology specialist consult, the echocardiogram confirmed the aortic ectasia (Figure 3), which was further analyzed with an angio-CT that showed a 17 mm aortic root (z-score +8.65), 9 mm ascending aorta (z-score +0.69), 7 mm transverse arch (z-score +0.67), 6 mm descending aorta (z-score +0.2), 5.3 mm diaphragmatic aorta (z-score +0.3), and 4 mm subrenal aorta. All major superior aortic branches were also tortuous (Figure 4).

The brain MR demonstrated a signal impairment in the para- and periventricular white matter, reduced thickness of the corpus callosum, and a slight bulge of the lateral ventricles. Lastly, a mild expansion of the periventricular spaces was found in association with a subarachnoid cyst (Figure 5).

The plastic surgery and otolaryngology consult were scheduled at two months to plan the corrective surgery for the cleft palate, and manipulations for the contractures of feet joints were started after the physiatrist evaluation. The ophthalmologist did not find any ocular features of the connective tissue disorders.

Chromosomal analysis on peripheral blood lymphocytes revealed a normal karyotype (46, XY) so a molecular genetic analysis through next-generation sequencing (NGS) for aortic anomalies was requested on the peripheral blood cells. This test revealed a c.1658C>A pathogenetic mutation of exon 8 of *TGFBR2.* The mutation was not detected in the parental DNA blood samples and confirmed the diagnosis of Loeys–Dietz syndrome in our proband.

The patient started a neurologic follow-up that documented a delay in gaining sitting position, a fix to follow moving objects, and reduced muscle tone at the inferior limbs. A mild improvement was achieved with psychomotricity. To prevent aortic dissection, the baby was initiated to medical therapy with beta-adrenergic blockers in addition to strict ultrasound monitoring of aortic and large vessel diseases. As of the writing of this report, the patient was 9 months old and no cardiovascular events had been recorded. On follow-up echocardiography, there was no increase in the aortic root at a dilation of 17 mm (z-score +8.65), while the aortic valve regurgitation was reduced. Correction surgery has not proved necessary for the time being. The child had optimal weight gain for his age with a BMI between the 50th and 75th percentile.

## 3. Discussion

We report a case of Loeys–Dietz syndrome type 2 due to a pathogenic variant in the *TGFBR2* gene associated with a cleft palate and aortic dilatation. The first mutations identified in the TGF-beta cascade concern the subunits of the TGF-beta receptor (*TGFBR1*, *TGFBR2*). These mutations are associated with a phenotype with a prevalence of craniofacial alterations and identify as the clinical variant LDS type 1 [5]. With the advent of new gene sequencing techniques, a genotype–phenotype correlation was sought. In 2012, two cases of patients with familial thoracic aortic aneurysm disorder due to pathogenic variants in *TGFBR2* were described [6]. These patients, while having features in common with Marfan syndrome, had unique features of LDS such as arterial tortuosity, clubfoot, hypertelorism, and bicuspid aortic valve. Therefore, patients with familial thoracic aneurysm disease with the genetic variant *TGFBR2* are referred to as LDS type 4 [7].

Although LDS presents with great phenotypic variability, a categorization on LDS genes is critical for the diagnosis and subsequent management of LDS patients. Cleft lip with or without cleft palate is one of the most common congenital malformations, occurring in about 1 in 600 to 700 births in the USA. A cleft palate alone is less common (1 in 1000 to 1500 births in the USA) but is more often associated with syndromes and not prone to prenatal diagnosis due to technological limitations [8]. In our case, the presence of a cleft in the soft palate of the newborn initiated the diagnostic investigation procedure. Careful evaluation of further abnormalities that emerged during hospitalization led us to suspect LDS. Thus began a multidisciplinary program of diagnosis, follow-up, and treatment. In the literature, different cases of LDS are described. In three of these, the suspicion of the syndrome came from the presence of large aneurysmal arteries. In the first case, an aneurysmatic aortic root was shown in a 19-week-old fetus who developed skeletal and craniofacial features of LDS after birth. Molecular analysis on the patient revealed a de novo mutation in *TGFBR2* [9]. Complex congenital heart disease, double-outlet right ventricle and interrupted aortic arch, with a huge aneurysm of the pulmonary arteries was found in a 36-week-old fetus who showed other features of LDS such as cleft of the soft palate, bifid uvula, and thin skin at birth. The baby unfortunately died the 21st day after birth from pulmonary arteries rupture following an attempt at unconventional clip banding. Additionally in this case, as in the previous one by Viassolo et al., pulmonary expansion from the fetal period led us to suspect a connective tissue disorder such as Marfan syndrome. There is no precedent in the literature of LDS associated with a congenital heart disease complex such as double-outlet right ventricle or interruption of the aortic arch. In such a case, the cardiovascular lesion as an expansion of the great vessels, that is, the aorta or pulmonary artery, may be aggravated during the fetal period. Consequently, the fetus may die in utero or shortly after birth due to the strong and continuous stress of the vessels. A post-mortem genetic analysis showed a de novo p.Thr200Pro (c.598A>C) mutation of *TGFBR1* [10]. A recent case of LDS, identified by the group from Trieste with Baldo et al., described a small male born at 38 + 1 weeks of gestation with hypotonia, joint hypermobility, arachnodactyly, and joint contractures of the fingers, as well as a senile and facial dysmorphism. Suspecting a connective tissue disorder, an echocardiography was performed, which also revealed a 13 mm dilation of the aortic root [11]. In this case, a trio-based whole-exome sequencing found a novel de novo variant in the *TGFBR2* gene. However, the same group identified a second case of LDS in a female born at 41 + 3 weeks of gestation. Unlike the previous case, the newborn presented a cleft palate and minor dysmorphisms but no alteration on the echocardiographic examination, although the genetic test confirmed the presence of a pathogenetic variant of *TGFB3* [12]. Yetman et al. described cases of newborns with genetically confirmed LDS. Each had slightly different musculoskeletal or cranial characteristics, but all related to connective tissue disease. In all cases, there was cardiac involvement represented by aortic enlargement with aortic valve insufficiency with or without patent ductus arteriosus, but the cardiac evolution was different. Two infants remained stable for years on beta-blocker or angiotensin-converting enzyme (ACE) inhibitor therapy, two required early aortic root surgery, and one died from aortic dissection following an urgent replacement of the aortic valve [13]. Muramatsu et al. reported on a newborn with long eyelid slits, a cleft palate, and mild retrognathia. The echocardiography showed a large ventricular septal defect (VSD), an atrial septal defect (ASD), a dilation of the aortic valve, and aneurysms of the aortic root and main pulmonary artery (PA). The baby developed congestive heart failure, so a PA banding was performed on day 12. Medical therapy with angiotensin-converting enzyme inhibitors was started to prevent vascular rupture and corrective surgery for a patch closing the VSD was performed at 2 months of age. The vascular dilatation remained stable up until 2 years of follow up. The genetic test identified a heterozygous T→A transition at nucleotide position 1370 in exon 5 of *TGFBR2* [13]. Individuals with LDS are likely to develop allergic diseases including asthma, food allergy, eczema, allergic rhinitis, and eosinophilic gastrointestinal diseases. In some affected subjects, high levels were evident of immunoglobulin E, eosinophil count, and cytokine T helper 2 (TH2) in the plasma [14,15]. Recently, Valenzuela et al. reported on another case of aortic sinus enlargement with a normal aortic annulus in a neonate with prominent arthrogryposis and a missense mutation in the coding sequence of *TGFBR2* [16]. In our case, the aortic root aneurysm associated with a small not hemodynamically significative patent ductus arteriosus and other congenital heart malformations or enlargement of pulmonary arteries were absent, suggesting great variability in the vascular tree anomalies in LDS syndrome. An important detail of our experience is that the finding of the aortic defect was accidental because, in the first days of the newborn’s life, there were neither signs nor symptoms of aortic dysfunction. The possible precocious vascular ruptures could justify preventive echocardiography in a newborn or even child with cleft of the soft palate or other signs of connective disorder and strict cardiology follow up or even early treatment [14]. In our case, it was not necessary to immediately perform a corrective surgery on the aorta; however, in the literature, a recent case of LDS type 2 associated with uvula bifida reports in detail on the multistep replacement of the aorta and subclavian artery in a 9-year-old child [17]. Finally, LDS patients need specialist neuropsychiatric consulting. While the multiple cysts in the frontal horn of both lateral ventricle and other germinolytic cysts in the subependymal area were already described [12] and seem to be benign and gradually shrink with the growth of the baby, neurological impairment is possible and explicated in our patient with generalized hypotonia that is benefitting from early psychomotricity.

## 4. Conclusions

The pleiotropic effect of genetic mutations associated with LDS makes the diagnosis more subtle. As has been widely reported, the characteristics of LDS may not be known from the fetal time or first days of life, so a careful family investigation can lead us to perform genetic investigations to determine a diagnosis of LDS.

Our case confirms the benefit of performing a careful examination of the patient in the neonatal resuscitation unit and in newborns with signs of connective disease to evaluate a possible association with a rare syndrome. Both specialistic consulting and molecular genetic diagnosis are imperative to confirm the primitive findings and begin a planned and tailored follow up and treatment program for LDS. Prenatal diagnosis for pregnancies at increased risk for LDS is possible if the disease-causing mutation in the family is known. In newborns with vascular anomalies, an early diagnosis is essential to ensure medical or surgical treatment that avoids lethal consequences. The possible benefits of beta-blockers or ARBs reported in the literature should encourage their immediate use when vascular anomalies are detected. The cardiovascular surgical approach should only be used in cases of nonresponsiveness to drug treatment.

Based on our clinical case, echocardiographic screening should be proposed at birth in all newborns with cleft palate to exclude aortic dilatation and possibly prevent short- and long-term consequences. We, ourselves, would not have arrived at the diagnosis so quickly without a screening echocardiogram.

We hope that new clinical trials will be conducted that identify genes and molecules useful in the medical therapy of LDS and limiting the progression of vascular abnormalities.

## Figures and Tables

**Figure 1 children-09-01290-f001:**
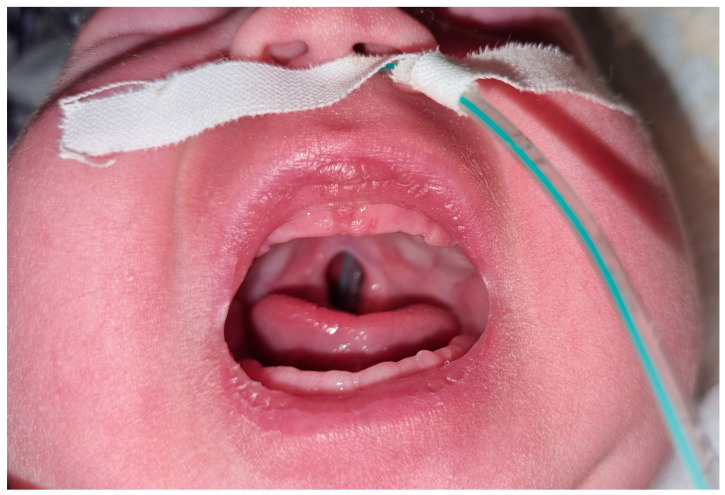
Cleft palate.

**Figure 2 children-09-01290-f002:**
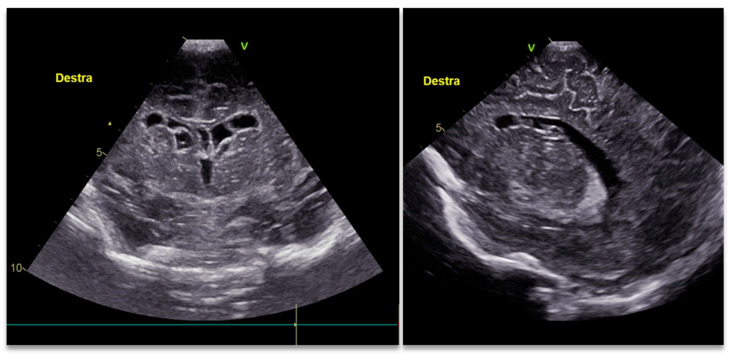
The cranial ultrasound shows gross cysts in the frontal horn of both lateral ventricles and small germinolytic cysts in the subependymal area. The former are probably cystic conversion of periventricular leukomalacia subsequently merged in the frontal horns (Destra stand for Right).

**Figure 3 children-09-01290-f003:**
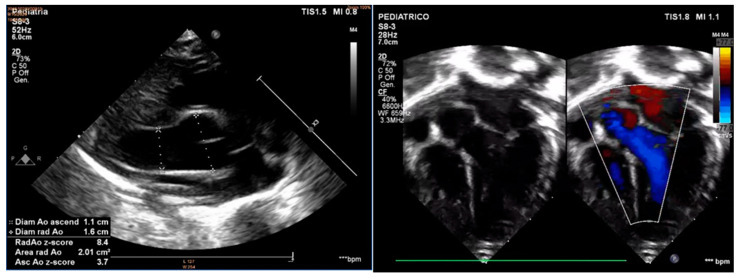
The echocardiography shows a wide dilatation of the aortic root with normal annulus and ascending aorta. The aortic valve was continent with preserved ventricular function and no additional intracardiac defects.

**Figure 4 children-09-01290-f004:**
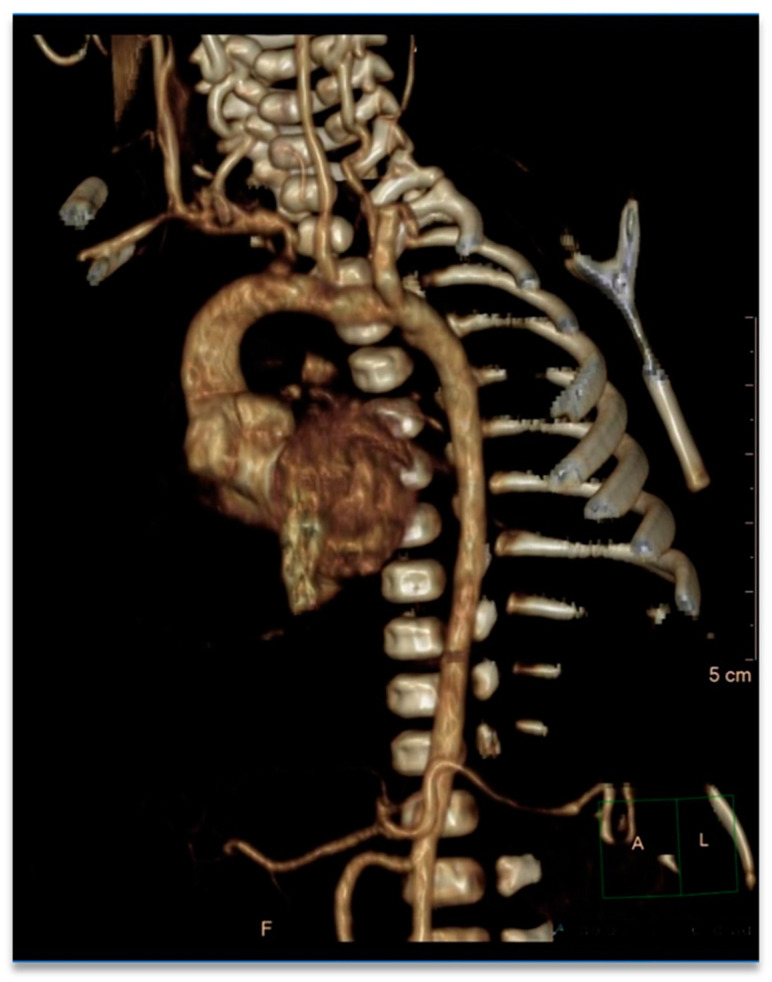
The angio-CT shows clearly an ectatic aortic root with an elongated transverse arch and descending aorta. The major superior aortic branches were tortuous (F stand for front).

**Figure 5 children-09-01290-f005:**
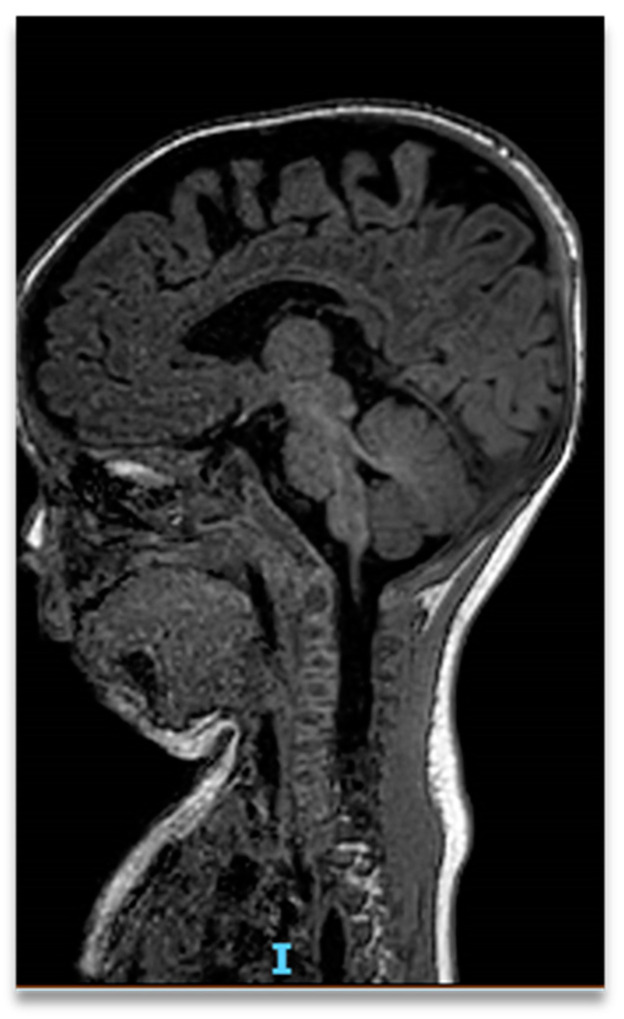
The brain MRI reveals a signal impairment in the para- and periventricular white matter, reduced thickness of the corpus callosum, and slight bulge of the lateral ventricles. A mild expansion of the periventricular spaces was found particularly in the temporomandibular region, with a contextual subarachnoid cyst.

## Data Availability

All data and material analyzed in this study are included in this published article.

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
