# Peer review of "Cleft Palate and Aortic Dilatation as Clues for Loeys–Dietz Syndrome"

_children, 2022, doi:10.3390/children9091290_

Round 1
Reviewer 1 Report (New Reviewer)
Congratulation to this nice work. The article is a case report on a patient with typical Loeys-Dietz-Syndrome type 2, one of a group of rare familiar aortic diseases. Very nice pictures show the typical features of patients with LDS type 2.
Study design:
Case report.
Comments:
Abtract:
Very good. Perhaps the listing of the genes should be put in the order of the LDS types 1 to 6.
Introduction:
Please do briefly introduce the six types of LDS in the introduction. Their differentiation is clinically important. Even patients harbouring TGFBR1 and TGFBR2 mutations, largely differ clinically. Therefore, it would be advisable not to seen as one diagnosisumped together.
Case report:
Very good. Very clear pictures of the typical features
Discussion:
In the discussion it is mentioned to screen all patients with palate cleft by echocardiography to exclude aortic dilatation. This is an excellent thought. This child’s diagnosis would not have been made without the echocardiogramm.
However, prophylactic Losartan therapy is not being discussed, but mandatory and often strikingly successful in preventing further aortic widening. Pediatric patients might actually grow out of their aortic dilatation.
In the discussion paragraph the text starts with a completely wrong introduction. This case is a child with type 2 LDS, mutation in TGFBR2! Possible spelling mistake, but the mentioned gene also exists…(see above).
From line 170 – line 213 other case reports from the literature are reported and discussed in comparison, which show mutations on different genes. That is misleading, since the clinical variation between genes might be huge. This holds true even between patients with mutations in the same gene. Therefore, please only discuss case reports describing patients with LDS type 2 also having a mutation on the TGFBR2 gene.
Conclusion:
Echocardiographic screening of all patients with cleft palate to exclude aortic dilatation should be recommended in the conclusion. Diagnosis of LDS is often delayed, e.g. this child’s diagnosis would not have been made without the echocardiogramm.
General comments, suggestion:
Though, this case report does not give any new information, it shows very beautifully an extremely typical patient with excellent pictures, therefore should be accepted with major revisions.
Author Response
Thanks for your comments.
Reviewer 1
Congratulation to this nice work. The article is a case report on a patient with typical Loeys-Dietz-Syndrome type 2, one of a group of rare familiar aortic diseases. Very nice pictures show the typical features of patients with LDS type 2.
Study design:
Case report.
Comments:
Abtract: Very good. Perhaps the listing of the genes should be put in the order of the LDS types 1 to 6.
Response: This correction was made (line 18).
Introduction: Please do briefly introduce the six types of LDS in the introduction. Their differentiation is clinically important. Even patients harbouring TGFBR1 and TGFBR2 mutations, largely differ clinically. Therefore, it would be advisable not to seen as one diagnosisumped together.
R: The different characteristics of LDS1-6 have been described and reported in the text (line 55-71).
Case report: Very good. Very clear pictures of the typical features
Discussion: In the discussion it is mentioned to screen all patients with palate cleft by echocardiography to exclude aortic dilatation. This is an excellent thought. This child’s diagnosis would not have been made without the echocardiogramm.
However, prophylactic Losartan therapy is not being discussed, but mandatory and often strikingly successful in preventing further aortic widening. Pediatric patients might actually grow out of their aortic dilatation.
In the discussion paragraph the text starts with a completely wrong introduction. This case is a child with type 2 LDS, mutation in TGFBR2! Possible spelling mistake, but the mentioned gene also exists…(see above).
From line 170 – line 213 other case reports from the literature are reported and discussed in comparison, which show mutations on different genes. That is misleading, since the clinical variation between genes might be huge. This holds true even between patients with mutations in the same gene. Therefore, please only discuss case reports describing patients with LDS type 2 also having a mutation on the TGFBR2 gene.
R: I apologize for the blunder at the beginning of the discussion. It was correct! Cases irrelevant to the case treated as LDS3 associated with another gene mutation were also removed.
Conclusion: Echocardiographic screening of all patients with cleft palate to exclude aortic dilatation should be recommended in the conclusion. Diagnosis of LDS is often delayed, e.g. this child’s diagnosis would not have been made without the echocardiogramm.
R: As suggested, the conclusions suggest echocardiography screening in all newborns with cleft palate at birth. As in our case, in fact, it can be essential for an early diagnosis of LDS.
General comments, suggestion: Though, this case report does not give any new information, it shows very beautifully an extremely typical patient with excellent pictures, therefore should be accepted with major revisions.
Please see the revised manuscript.

Reviewer 2 Report (New Reviewer)
Dear Authors
LDS is still a new disease with continous discoverage of new aspects of the disease. You precisely describe all patients symptoms a well present the whole case report.
However, cleft palate and aortic dilatation are already well known manifestations of the disease. What is essentially new in your report? Please point this out.
As far as I know, patients with TBFBR2 mutations are classified as LDS type 2, and not type I. Why did you classify your patients as type I?
Your patients has several vascular manifestations (aortic root aneurysm, turtous supraaortic arteries and aortic arch). You correctly describe, that clinical progression of aortic aneurysm or dissection in LDS patients show a rapid progression. Despite the medial therapy you prescribe your patients, he will probably require operative repair of the aorta and its majro supraaortic branches. You should highlight this aspect (especially as you have the aortic dilatation in your title already) and discuss it with relevant literature such as a recently published case report on LDS type II with also uvula bifida and aortic manifestation requiring extensive multiple-staged replacement of the aorta and its supraaortic branches (Dueppers et al. Complex Multi-Stage Total Aortic and Subclavian Artery Replacement in a 9-year old boy with Loeys-Dietz-Syndrome, AVS 2022.)
Author Response
Thanks for your comments.
Reviewer 2
Dear Authors
LDS is still a new disease with continous discoverage of new aspects of the disease. You precisely describe all patients symptoms a well present the whole case report.
However, cleft palate and aortic dilatation are already well known manifestations of the disease. What is essentially new in your report? Please point this out.
R: As underlined in the conclusions, our case wants to underline the importance of making the diagnosis of LDS early in order to prevent its consequences, especially cardiovascular ones. In this sense, it is suggested the importance of carrying out, as in our case, an echocardiographic screening at birth in all newborns with cleft palate.
As far as I know, patients with TBFBR2 mutations are classified as LDS type 2, and not type I. Why did you classify your patients as type I?
R: Yes, I apologize it was a mistake. It was correct.
Your patients has several vascular manifestations (aortic root aneurysm, turtous supraaortic arteries and aortic arch). You correctly describe, that clinical progression of aortic aneurysm or dissection in LDS patients show a rapid progression. Despite the medial therapy you prescribe your patients, he will probably require operative repair of the aorta and its majro supraaortic branches. You should highlight this aspect (especially as you have the aortic dilatation in your title already) and discuss it with relevant literature such as a recently published case report on LDS type II with also uvula bifida and aortic manifestation requiring extensive multiple-staged replacement of the aorta and its supraaortic branches (Dueppers et al. Complex Multi-Stage Total Aortic and Subclavian Artery Replacement in a 9-year old boy with Loeys-Dietz-Syndrome, AVS 2022.)
R: Thanks for the comment.
The request was entered in the text (line 147-150).
Please see the revised manuscript.
Reviewer 3 Report (New Reviewer)
The authors present the detailed description of the ultra-rare disease: Loeys-Dietz syndrome.
The very precise depiction of the rare disease presents the enormous substantive value for the neonatologists and pediatricians.
The article also points out that the children who are born with the cleft palate or bifid uvula should be given further attention and evaluation. As the mentioned abnormality can be regarded as a common inborn anomaly, the underlying cause of it such as a rare disease should always be taken under consideration.
For the full case report evaluation of the patient, in the section 2, the information about the follow up echocardiography after 9 months should be stated. This should include the measurments of the particular aortic parts in milimeters and it should be presented together with the z- score scale for height and body mass. As well other information about the cardiac abnormalities such as valves insufficiency should be given like it was presented at the beginning of the article.
I recommend this article for publication.
Author Response
Thanks for your comments.
Reviewer 3
The authors present the detailed description of the ultra-rare disease: Loeys-Dietz syndrome.
The very precise depiction of the rare disease presents the enormous substantive value for the neonatologists and pediatricians.
The article also points out that the children who are born with the cleft palate or bifid uvula should be given further attention and evaluation. As the mentioned abnormality can be regarded as a common inborn anomaly, the underlying cause of it such as a rare disease should always be taken under consideration.
For the full case report evaluation of the patient, in the section 2, the information about the follow up echocardiography after 9 months should be stated. This should include the measurments of the particular aortic parts in milimeters and it should be presented together with the z- score scale for height and body mass. As well other information about the cardiac abnormalities such as valves insufficiency should be given like it was presented at the beginning of the article.
R: Thanks for the comment. This information was inserted into the text (line 163-168)
I recommend this article for publication.
Please see the revised manuscript.

This manuscript is a resubmission of an earlier submission. The following is a list of the peer review reports and author responses from that submission.
Round 1
Reviewer 1 Report
Zaza et al describe a case of Loeys Dietz syndrome presenting with cleft palate and aortic root dilation. The case report requires improvement. The text seems to wander and should be tightened to focus on the pertinent clinical features (rather than the list of absent clinical characteristics). In addition, as the authors point out, this topic and its presentation has been reviewed at length in other publications. The presentation of this case is not particularly novel.
Reviewer 2 Report
Please see the attached review.
